# Assemble the shallow or integrate a deep? Toward a lightweight solution for glyph-aware Chinese text classification

**Jingrui Hou**[1], **Ping Wang**[2,3]*

1 Department of Computer Science, School of Science, Loughborough University, Loughborough, Leicestershire, United Kingdom, 2 Center for the Studies of Information Resources, Wuhan University, Wuhan, Hubei, China, 3 School of Information Management, Wuhan University, Wuhan, Hubei, China

* wangping@whu.edu.cn

## Abstract

As hieroglyphic languages, such as Chinese, differ from alphabetic languages, researchers have always been interested in using internal glyph features to enhance semantic representation. However, the models used in such studies are becoming increasingly computationally expensive, even for simple tasks like text classification. In this paper, we aim to balance model performance and computation cost in glyph-aware Chinese text classification tasks. To address this issue, we propose a lightweight ensemble learning method for glyph-aware Chinese text classification (LEGACT) that consists of typical shallow networks as base learners and machine learning classifiers as meta-learners. Through model design and a series of experiments, we demonstrate that an ensemble approach integrating shallow neural networks can achieve comparable results even when compared to large-scale transformer models. The contribution of this paper includes a lightweight yet powerful solution for glyph-aware Chinese text classification and empirical evidence of the significance of glyph features for hieroglyphic language representation. Moreover, this paper emphasizes the importance of assembling shallow neural networks with proper ensemble strategies to reduce computational workload in predictive tasks.

## Introduction

Text classification is a common task in natural language processing (NLP), and most studies focus on English corpora [1]. However, some hieroglyphic languages, such as Chinese, have unique characteristics that differ from alphabetic languages. As a result, Chinese NLP researchers have begun to use the internal features of Chinese ideogram characters for semantic enhancement, proposing various approaches. During the early stage, the character-enhanced approach is widely used in Chinese word embedding [2]. Some studies have concentrated on Chinese character semantics at a finer granularity than full words [3–6]. Additionally, some researchers have regarded Chinese characters as images, and vision technologies have even been used to extract semantic information [7–9]. With the advent of pretrained language models (PLMs) [10], the internal glyph or phonetic features were added to PLMs [11, 12].

**Data Availability Statement:** The domain classification dataset employed in this study can be accessed at "http://thuctc.thunlp.org", while the sentiment classification dataset utilized is available

at "https://github.com/SophonPlus/
ChineseNlpCorpus".

**Funding:** This work was financially supported by
the National Natural Science Foundation of China
under grant number 72074171. The funder did not
participate in the study design, data collection and
analysis, decision to publish, or in the preparation
of the manuscript.

**Competing interests:** NO authors have competing
interests.

Chinese text representation methods that take into account the individual characters or
glyphs can be referred to as "glyph-aware Chinese text representation". One of the most com-
mon applications of these technologies is text classification, which is a fundamental and essen-
tial task in NLP and data science [13]. However, research questioning the use of complex
architectures in text classification tasks has never ceased, as complex networks require power-
ful computing capacity and are not friendly to machines with limited computation capacity
[14–16]. Moreover, a recent survey [17] has shown through a comprehensive experiment with
diverse text classifiers and datasets that shallow networks are less powerful than complex ones.
As a result, a contradiction exists between the high performance and lightweight nature of text
classifiers, making it challenging to overcome this dilemma.

Ensemble learning, particularly ensemble learning of shallow networks, offers a new solu-
tion to the dilemma of balancing high performance and lightweight text classifiers. By combin-
ing several weak classifiers in a specific manner, a stronger classifier can be obtained, thereby
enhancing model generalizability [18], minimizing overfitting [19], and improving imbal-
anced learning [20]. The ensemble of shallow networks has been demonstrated to be more
effective than deeper neural networks in various scenarios, including fake reviews classification
[21] and Wikipedia quality classification [22]. Inspired by these studies, this paper aims to
design a lightweight solution for glyph-aware Chinese text classification using ensemble learn-
ing. To achieve this, we propose the following research question:

*RQ1: What are the key considerations and design choices for developing a lightweight method
that combines ensemble learning and shallow neural networks for glyph-aware Chinese text
classification?*

There are several factors to consider when designing the ensemble learning approach for
glyph-aware Chinese text classification. The first is to select appropriate base learners and a
meta-learner [18]. The second is to efficiently utilize glyph features and choose the combina-
tion strategies of text features and glyph features. Lastly, the ensemble approach should avoid
excessive computational requirements. Once the first question is addressed, it will be necessary
to compare its performance to other state-of-the-art classifiers with deeper architectures. It is
worth noting that sentiment and domain classification are two main scenarios in text classifi-
cation [23]. Therefore, we present the second research question:

*RQ2: To what extent can a lightweight method outperform deep neural models in sentiment and
domain classification scenarios for Chinese text?*

This study also aims to investigate how glyph features function in the ensemble learning
approach and how they can enhance classification performance. Additionally, not every
module carries equal importance in ensemble learning, and some may have varying impacts
[24, 25]. It is necessary to examine how each module influences the overall approach and
identify the most lightweight solution. Therefore, the following research questions are
proposed:

*RQ3: What is the impact of glyph features on the accuracy and robustness of Chinese text classifi-
cation models?*
*RQ4: What practical solution yields the best performance for glyph-aware Chinese text classi-
fication in terms of efficiency and efficacy?*

The remainder of this paper is organized as follows. Section 2 reviews related research. Sec-
tion 3 presents the proposed method for RQ1. Sections 4 to 6 report the results and analysis
for RQ2 to RQ4. Section 7 discusses the findings and implications. The final section provides a
concise conclusion and outlines future research directions.

## Related works

### Research on Chinese text representation

Even recently, a considerable number of researchers have inclined towards solving Chinese text classification without considering internal features within Chinese texts and have focused on complex deep network architectures [26, 27]. However, the application of Chinese representations using glyph features has been a long-standing research topic. To make full use of the word-character-radical composition of Chinese characters, Shi et al. [4] proposed radical embedding trained by the word2vec algorithm. To utilize the semantic information of radicals, Yin et al. [6] presented a multigranularity embedding for Chinese words, incorporating finer-grained semantics from characters and radicals to enrich the word embeddings. Sun et al. [28] proposed a dedicated neural architecture with a hybrid loss function and integrated radical information by performing a softmax operation on each character. Li et al. [5] introduced a component-enhanced Chinese character embedding model based on finer-grained semantics associated with the components of characters, and the benefits of considering radicals when learning rich semantic representations have been verified by their experimental results. Chen and Hu [3] proposed a model to extract the intrinsic information in the Chinese corpus with conversion and radical escaping mechanisms. Tao et al. [29] introduced a radical-aware attention-based model to jointly leverage features of Chinese characters, words, character-level radicals, and word-level radicals.

Radical features have even been regarded as visual features. Liu et al. [7] created Chinese embeddings using the images of every Chinese character, where radicals were used as shared information to improve character embedding. Dai and Cai [8] rendered Chinese glyphs into raw pixels and created a glyph-aware embedding for Chinese. Similarly, Su and Lee [9] employed a convolutional autoencoder model to learn bitmaps of characters to enhance word representation with character glyphs. The authors concluded that the convolutional neural network (CNN) is suitable for visual radical modeling. Xuan et al. [30] proposed a fusion glyph network where character representation and visual glyph information are combined using a fusion mechanism.

After the tremendous success of PLMs, internal features of Chinese characters started being utilized in Chinese-based PLM research. Meng et al. [11] employed historical Chinese scripts to enrich the pictographic evidence in Chinese characters, and the resulting embedding was concatenated with BERT to obtain a comprehensive glyph representation. Later, Sun et al. [12] proposed ChineseBERT, where glyph and pinyin information of Chinese characters are inputted into language model pretraining. Additionally, Chinese PLMs incorporated multiple glyph features generated by different split strategies, along with pinyin embedding [31].

### Ensemble learning in text classification

Ensemble learning has been widely applied in text classification scenarios due to its ability to yield accurate predictions [18]. Recent studies have demonstrated the effectiveness of ensemble learning in text classification. Huang and Chen [32] applied a self-adaptive harmony search algorithm to optimize ensemble learning and achieved the highest accuracy in fake news detection. Hou et al. [22] used an ensemble of sentence embedding and statistical features to address Wikipedia quality classification. Wang et al. [33] even applied ensemble learning in the legal field, and their results showed that it could provide a reliable decision basis in detecting arson cases. Abbasi et al. [34] proposed an approach for authorship identification that combined PLMs and traditional machine learning classifiers. Kazmaier and van Vuuren [35] compared various ensemble learning configurations for sentiment analysis and proved the effectiveness

of ensemble learning in text classification. Li et al. [36] proposed an ensemble method called fastEmbed that combines the fastText and LightGBM algorithms to predict the exploitability and exploitation of vulnerabilities on unbalanced datasets. The method extracts key features from the vulnerability-related text, which is considered the most important feature of the model.

## RQ1: Designing a lightweight ensemble learning solution for glyph-aware Chinese text classification

In this section, we outline the steps for building a lightweight glyph-aware Chinese text classification model (LEGACT).

### Efficient utilization of glyph features

A recent study [37] has shown that word segmentation is not necessary in deep learning tasks for Chinese text. Therefore, we use character sequences instead of word sequences for initial text representation. To utilize glyph features, we focus on the use of "radicals", which are partial glyphs that convey semantic information [9]. Furthermore, while there are approximately 3,500 commonly used Chinese characters, only about 200 radicals are repeatedly used in different Chinese characters.

Typically, each Chinese character has exactly one unique radical. Consequently, there is a many-to-one relationship between characters and radicals. To acquire the radicals of each input Chinese character, we constructed an offline Chinese Radical Lexicon database from the online Xinhua Dictionary (https://zidian.aies.cn/), which is a reputable Chinese dictionary. Given a Chinese character sequence, we can easily map it into a radical sequence using the offline Chinese Radical Lexicon. For instance, the extracted radical sequence of the Chinese sentence "她想吃晚饭" is "女心口日饣". Table 1 provides detailed meanings for each character and radical. Once we have obtained the character and radical sequences, we can easily embed them into character-level and radical-level embeddings.

### Base learners

In fact, many sophisticated networks are constructed by deep stacking feedforward neural networks (FNN), convolutional neural networks (CNN), and recurrent neural networks (RNN) [38]. In our study, we selected a DNN (Dense neural network, consisting of one feedforward layer), a TextCNN [39], and BI-GRU (Bi-Directional Gated Recurrent Unit, one model in the RNN family) [40] as base learners to encode character features and radical features. Additionally, FastText [41], a fast and effective text classification tool that enhances input sequences of neural networks using n-gram word features, was also incorporated into the base learners.

As a shallow neural network is commonly defined as an architecture with only one hidden layer [42], we maintain this setting in our study. Additionally, since one-layer convolution networks are often not adequate for encoding text features, we use a three-layer module (shallow-and-wide CNN layer [39]) instead.

The details for each type of base learner are illustrated in Fig 1.

**Table 1. A Chinese character sequence and its corresponding radical sequence.**

|  | Character sequence | Radical sequence |
| --- | --- | --- |
| Chinese character | 她 想 吃 晚 饭 | 女 心 口 日 饣 |
| English equivalent | she, want(s) to, eat, dinner | female, mind, mouth, day, food |

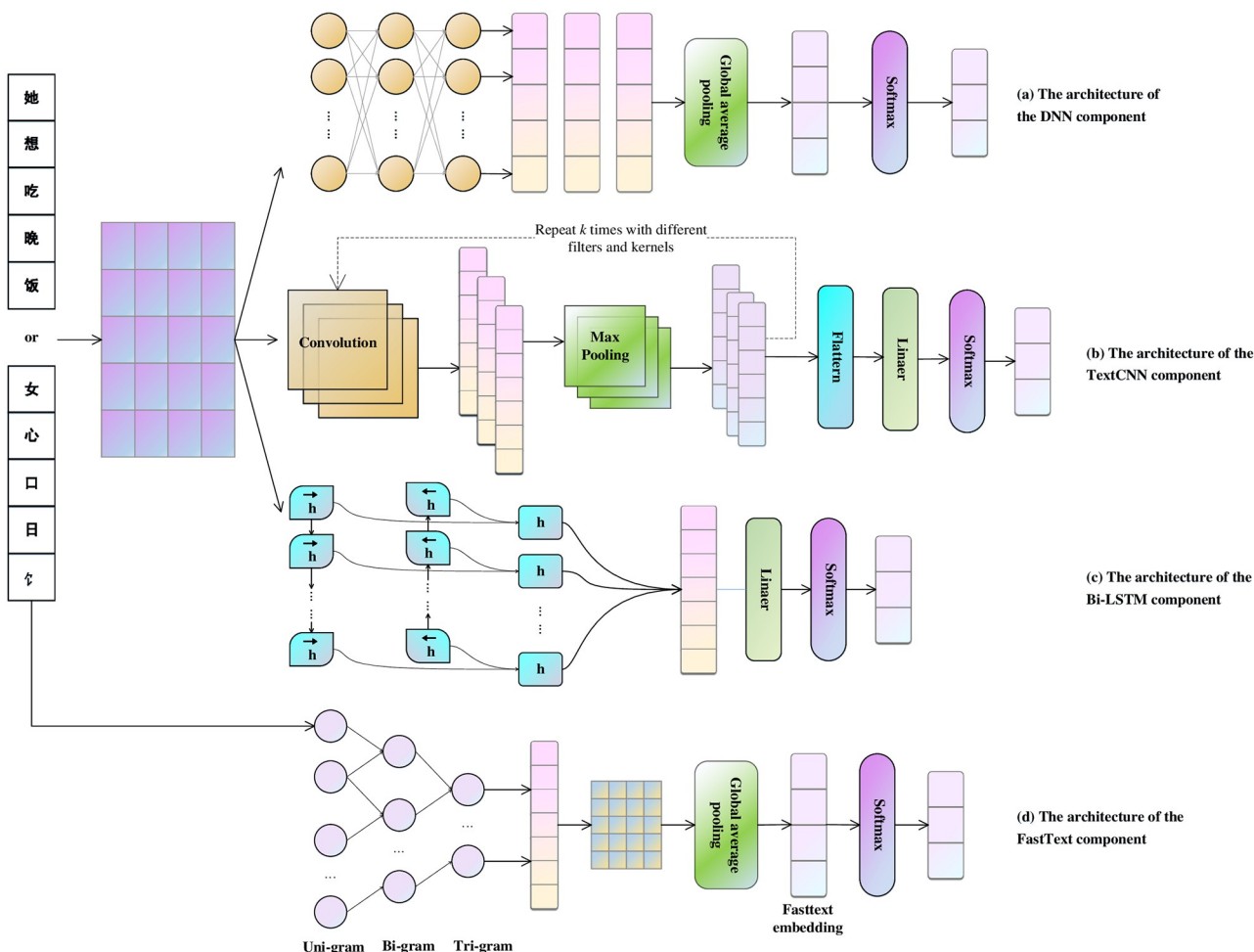

**Fig 1. Base learner architecture in LEGACT.** The depicted Deep Neural Network (DNN) comprises a hidden layer bridging the input and output layers. It utilizes a Global Average Pooling layer for encoding tensor dimensionality reduction. The TextCNN is structured with three sets of convolution and max-pooling layers for input vector encoding, and its output is subsequently flattened into a one-dimensional vector purposed for classification. The Bidirectional Gated Recurrent Unit (BI-GRU) is a sequential model featuring two GRU layers operating in forward and backward directions. FastText organizes the input into uni-gram, bi-gram, and tri-gram features, and the resulting embedding vectors are then fed into a global average pooling layer.

DNN, TextCNN, and GRU utilize a shared model input and embedding. The DNN segment employs multiple hidden layers, connected by fully connected layers, to process the character or radical-level embedding matrix. It is followed by a global max pooling layer that normalizes sequence lengths and extracts the maximum value from the hidden layer output. The TextCNN operates on the embedding matrix through a cyclic operation involving a one-dimensional convolution layer and a max pooling layer. The convolution layer captures semantic features using a sliding convolution kernel, while the max pooling layer reduces parameter size and ensures a consistent input length for subsequent cycles. After convolution and pooling, the two-dimensional vectors are flattened, and a batch normalization layer is added for smoothing. The BI-GRU structure learns the semantic representation by considering both forward and backward text sequences. The final representation is obtained by concatenating the two semantic encodings obtained after updating all elements in the sequence. FastText, despite its simplistic network structure, enriches the input with n-gram

features. These features, including uni-, bi-, and trigram aspects, are combined to form a new sequence. This sequence is then embedded and passed through a global average pooling layer to enhance sentence representation.

### Stacking ensemble and meta-learner

To prevent overfitting during ensemble learning, we use stacking ensemble to combine all the base learners with different data subsets. Using fixed fold IDs, the training set is divided into five subsets, ensuring that there are no duplicate samples between subsets. For each base learner, we train five models, using four subsets for training and one subset for validation. The different models have different validation sets, and the output logits vectors from each validation set are concatenated to create a new training set, which is used as input for ensemble learning. For the testing set, we use five models to predict the original testing set and obtain five logit vectors. Then, we apply an average function to calculate the mean logit values for each category. The resulting vectors are used as the prediction set for ensemble learning. To better demonstrate the proposed stacking ensemble process, we provide the following algorithm description, shown in Fig 2.

Once the process is complete, the four character-level and radical-level logit vectors are input into the meta-learner XGBoost [43]. XGBoost uses a decision tree to fit the last prediction residual and improves the model's performance through iteration. Using XGBoost to integrate network-based base learners has been proven to be an efficient and effective meta-learner in many studies [22, 44]. The process of using XGBoost with a stacking ensemble strategy is illustrated in Fig 3. It results in a final prediction by integrating character features and radical features produced by several parallel shallow neural networks.

## RQ2: Comparing the proposed lightweight method with other sophisticated methods that integrate deeper architectures

To address RQ2, we selected two datasets for Chinese text classification tasks, specifically domain classification and sentiment classification. Additionally, we chose strong baselines that have achieved state-of-the-art results in different periods.

### Dataset

The dataset for domain-based classification is called *THUCNews* [45]. It consists of real-world online news samples collected from the internet. We selected ten domains, with each domain containing 5,000 samples, to form our dataset. Another dataset we utilized is *Yf_amazon* [46], which is an online customer review dataset collected from Amazon China. This dataset includes three sentiment labels: "Negative", "Neutral", and "Positive". To ensure a balanced data distribution, we randomly selected 20,000 samples for each sentiment label.

Table 2 provides an overview of the corpus information used in this study.

### Baselines

To verify the performance of the proposed method, we adopted some multiple-neural-layer models achieving state-of-the-art performance in the early years, general PLMs and Chinese-oriented PLMs integrating glyph and phonetic features as baselines in this experiment. Multiple neural layer models include BI-GRU attention [47], CNN-LSTM [48], RCNN [49], and HAN [50]. General PLMs include RoBERTa [51]. Chinese-oriented PLMs include glyphBERT [11] and ChineseBERT [12]. In addition, we added an extra voting method in this experiment

---

**Algorithm**: Stacking ensemble process

**Input:**

 Original dataset $D = X, Y = \{(x_1, y_1), (x_2, y_2), \ldots, (x_m, y_m)\}$;

 Folds $k$;

 Base learners $W = \{l_1, \ l_2, \ldots, \ l_n\}$;

**Output:**

 Training set for ensemble learning $\Theta$

 Test set for ensemble learning $\Phi$

**Process:**

1: Split $D$ into $Train$ and $Test$;

2: Split $Train$ into $k$ disjoint equal-sized subsets: $s_1, \ s_2, \ldots, \ s_k$;

3: **for** $i \leftarrow 1 \ to \ n$ **do**:

4: **for** $j \leftarrow 1 \ to \ k$ **do**:

5: Test set:te $\leftarrow s_j$

6: Training set:tr $\leftarrow (Train - s_j)$

7: Train base learner with tr: $model_i \leftarrow train(l_i, \text{tr})$

8: Make predictions on te:$\theta_{ij} \leftarrow predict(model_i, \text{te})$

9: Make predictions on $Test$:$\varphi_{ij} \leftarrow predict(model, Test)$

10: **end for**

11: $\theta_i \leftarrow concatenate(\theta_{i1}, \theta_{i2}, \ldots, \theta_{ik})$

12: $\varphi_i \leftarrow average(\varphi_{i1}, \varphi_{ij}, \ldots, \varphi_{ik})$

13: **end for**

14: $\Theta \leftarrow \theta_1 \cup \theta_2 \cup \ldots \cup \theta_n$

15: $\Phi \leftarrow \varphi_1 \cup \varphi_2 \cup \ldots \cup \varphi_n$

16: **return** $\Theta, \ \Phi$

---

**Fig 2. Stacking ensemble process of in LEGACT.**

to validate whether the parallel combinations of simple networks are as efficient as the networks with deep architecture typically used for text classification.

## Performance comparisons

The proposed method integrates only four shallow neural networks and a machine learning classifier, allowing the experiments to be conducted on a laptop with an Intel 10750H CPU and 16 GB RAM. In contrast, the experiments for the other baseline models were conducted on a GPU workstation equipped with an RTX 3080 GPU. As shown in Table 3, the proposed method outperforms all traditional deep models (No.1-4) in both domain-based and sentiment-based classification tasks. It achieves very close performance to pretrained models in domain classification and stands out as the only method to achieve an accuracy over 0.80 for sentiment-based classification.

 Since text classification is not a complicated task, we can use model parameters as a representation of model complexity. Among the four parallel base learners, DNN, TextCNN,

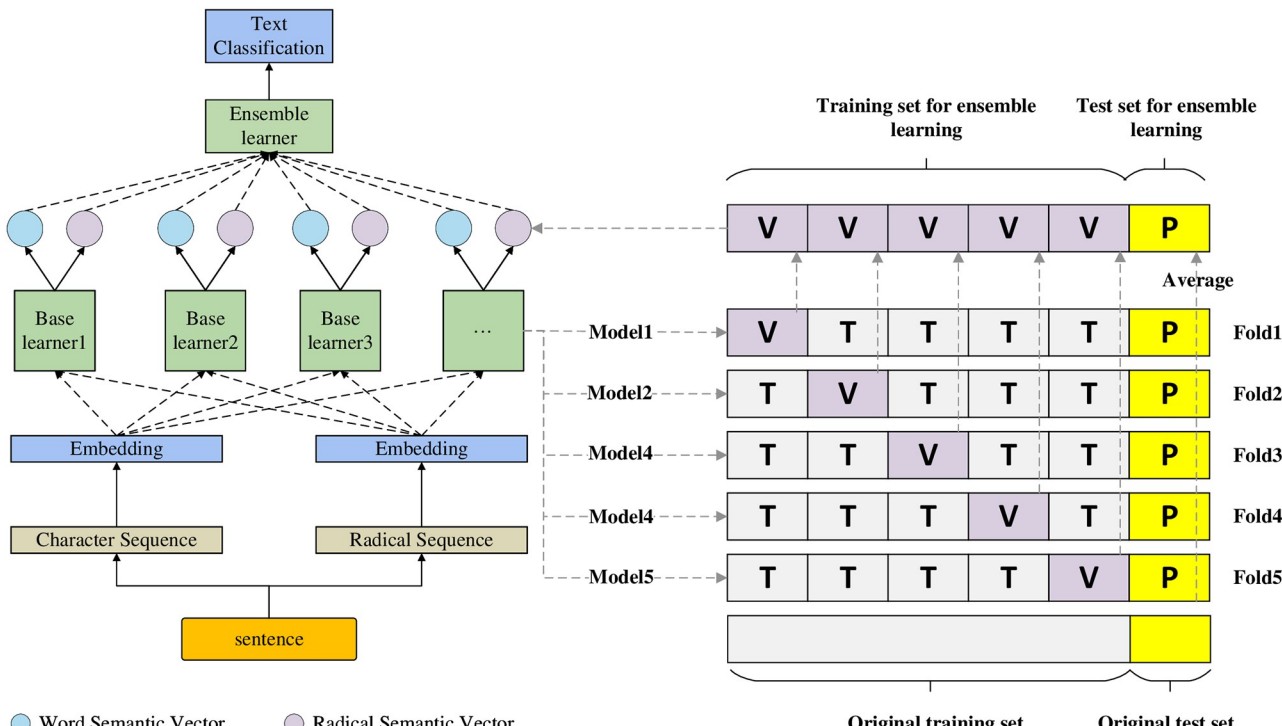

**Fig 3. Proposed ensemble approach for LEGACT.** "T", "V", and "P" represent training, validation and prediction sets, respectively. The process involves inputting the logit vectors from the character-level and radical-level base learners into the meta-learner XGBoost to obtain the final prediction.

**Table 2. Corpus information.**

| Number | name | Language | Number of samples | Number of categories | Characters per sample | Total distinct radicals | Total distinct characters |
|---|---|---|---|---|---|---|---|
| 1 | THUCNews | Chinese | 50,000 | 10 | 829 | 231 | 5,738 |
| 2 | Yf_amazon | Chinese | 60,000 | 3 | 190 | 253 | 6,798 |

**Table 3. Accuracy scores on domain-based and sentiment-based text classification.**

| Index | Model Name | Features used | Training device | Parameters (million) | Accuracy on domain classification | Accuracy on sentiment classification |
|---|---|---|---|---|---|---|
| 1 | BI-GRU Attention | Characters | GPU | 1.14 m | 0.953 | 0.759 |
| 2 | CNN-LSTM | Characters | GPU | 1.09 m | 0.939 | 0.778 |
| 3 | RCNN | Characters | GPU | 2.39 m | 0.952 | 0.713 |
| 4 | HAN | Characters | GPU | 2.48 m | 0.934 | 0.737 |
| 5 | RoBERTa | Characters | GPU | 10.31 m | **0.978** | 0.794 |
| 6 | GlyceBERT | Characters, glyph | GPU | 7.58 m | 0.976 | 0.756 |
| 7 | ChineseBert | Characters, glyph, pinyin | GPU | 14.73 m | 0.969 | 0.774 |
| 8 | Voting | Characters | CPU | 1.01 m, 2.24 m, 1.12 m, 2.00 m] | 0.965 | 0.797 |
| 9 | LEGACT(Our method) | Characters, radicals | CPU | 1.01 m, 2.24 m, 1.12 m, 2.00 m] | 0.973 | **0.806** |

BI-GRU, and FastText, they have 1.01, 2.24, 1.12, and 2.00 million parameters, respectively. These values are significantly lower than the pretrained models (No.5-7) shown in Table 3. However, the ensemble of these base learners demonstrates equivalent or even superior performance. Furthermore, it is worth noting that the voting method, which uses only character sequences as input and combines the predictions of the four base learners, also achieves decent accuracy scores. This result demonstrates that for Chinese text classification tasks, a voting method that incorporates several base learners may yield better performance compared to deeper networks.

## RQ3: Analysing how Chinese radical features impact the classification performance

To address RQ3, in this section, we further analyze the importance of including radical sequences in Chinese text classification. First, we input the character and radical sequences separately into each base learner to train individual classifiers. Then, a character-based classifier and a radical-based classifier for a base learner are combined for ensemble learning; this could be regarded as a simplified version of the proposed method using only one base learner. We conducted two groups of experiments, one for domain-based text classification and one for sentiment-based classification.

### Domain-based text classification experiment

The overall performance of the tested models is listed in Table 4. It is worth noting that although the Chinese radical lexicon contains fewer than 300 tokens, all the base learners can fit properly when using only radical features as input, and they achieve good performance on a classification task with 10 categories. In particular, when given only radical sequences as input, TextCNN and BI-GRU achieve accuracies over 0.9, while FastText and DNN achieved accuracies above 0.8 and 0.7, respectively.

Based on this comparison of the base learner performance under different inputs, we find that after combining the radical features, the accuracies of all the base learners improved: the DNN improved by 0.02, the TextCNN improved by 0.02, BI-GRU improved by 0.019, and FastText improved by 0.003. These results indicate the efficiency of applying radical features to the Chinese text classification task. To further investigate the fine-grained effects of incorporating radical features on the model, we drew confusion matrices based on the results of the four base learners, as shown in Fig 4.

Fig 4A–4C consist of three confusion matrices for DNN models provided with character, radical, and combined features separately as input. In the matrices, the cells on the diagonals (from upper-left to lower-right) denote the probabilities that the predicted labels match the true labels, whereas the other cells represent the probabilities of false predictions. We find that in the matrix (Fig 4B), the radical features do not yield particularly high accuracy, but they do

**Table 4. Base learner accuracy on the domain-based text classification.**

| Number | base learner | accuracy | | |
|---|---|---|---|---|
| | | character only | radical only | combined |
| 1 | DNN | 0.926 | 0.728 | 0.946 |
| 2 | TextCNN | 0.935 | 0.907 | 0.955 |
| 3 | BI-GRU | 0.943 | 0.918 | 0.962 |
| 4 | FastText | 0.962 | 0.901 | 0.965 |

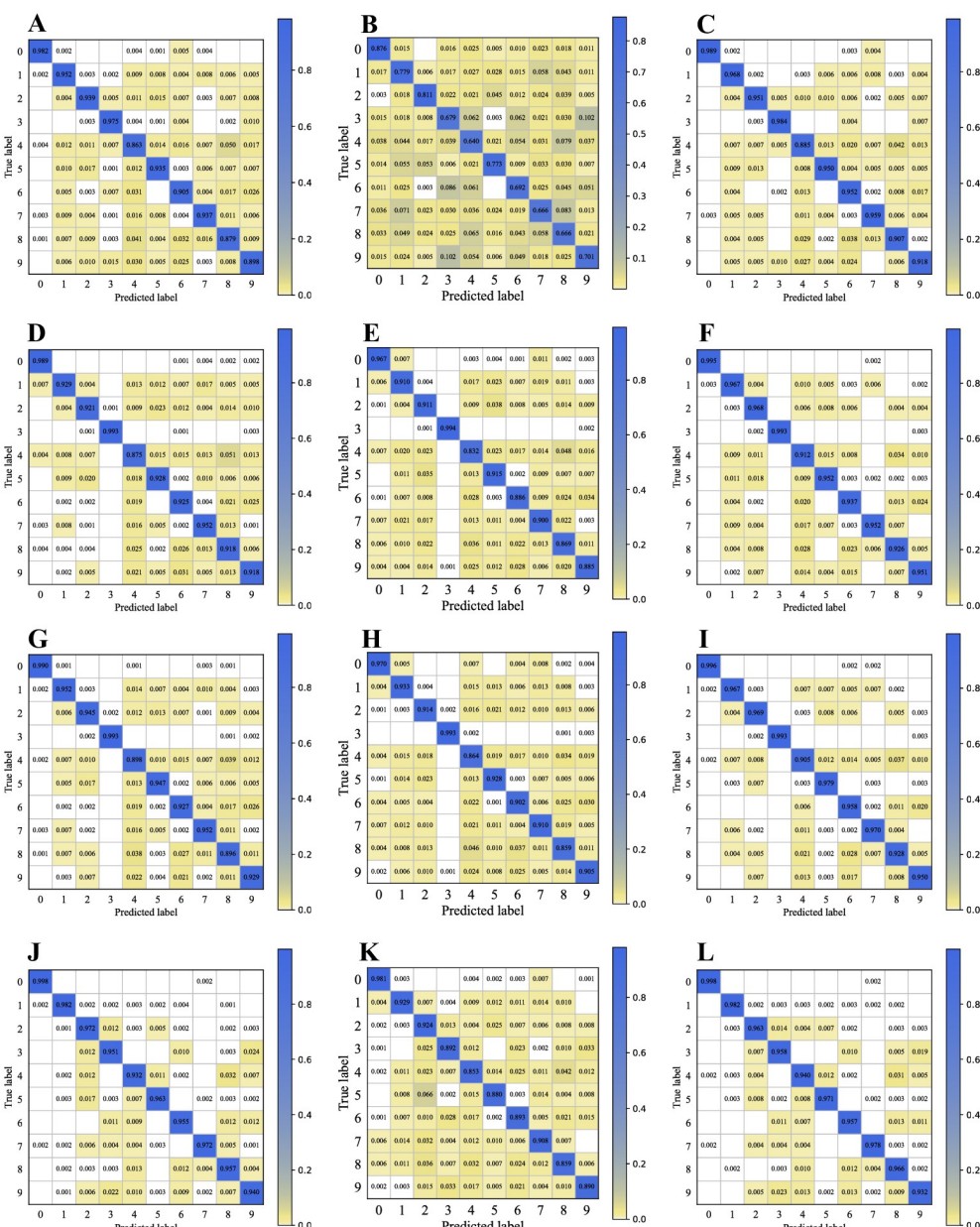

**Fig 4. Domain–based classification confusion matrices.** The results for different methods are shown: DNN with (A) character, (B) radical, (C) combined features; TextCNN with (D) character, (E) radical, (F) combined features; BI-GRU with (G) character, (H) radical, (I) combined features; FastText with (J) character, (K) radical, (L) combined features.0-Sports, 1-Entertainment, 2-Home Furnishing, 3-House Property, 4-Education, 5-Fashion, 6-Government, 7-Games, 8-Science & Technology, 9-Finance.

help to avoid some false predictions generated by character sequences. The accuracies in the matrix (Fig 4C) are overall higher than those in the matrix (Fig 4A). Comparing the cells on the diagonal lines of Fig 4A and 4C, we further notice that all the cells improve by different ranges.

Fig 4D–4F illustrate the performance of TextCNN given the same three types of input. Similarly, we find that the accuracy of each category on the diagonal increases as a whole from the

matrix (Fig 4D) to the matrix (Fig 4F). In this group, only the accuracy scores for "House Property" and "Games" remain unchanged. The accuracy of using radical sequence on "House Property" reached 0.994, which slightly exceeded the accuracy of the character sequence and the combined sequence for that same class. In addition, we found that the incorporation of radical features improves the model's performance by compensating for mispredictions caused by character sequences alone. Taking the first column in Fig 4D and 4F as an example, we find that 0.7% of samples in "Entertainment", 0.4% in "Education" and "Science & Technology", and 0.3% in "Games" are wrongly predicted as "Sports"; however, after the correction of the radical sequence, the first error rate is reduced to 0.003, and the latter errors are largely eliminated.

The matrices for the BI-GRU are shown in Fig 4G–4I. Comparing ten pairs of cells in matrix (Fig 4G) and in matrix (Fig 4I), it can be seen that all the accuracies in Fig 4I are higher than the corresponding accuracies in Fig 4G. In addition, the number of yellow cells in Fig 4I is smaller than that in (Fig 4G), and their colors are lighter. Such comparisons show the effect the radical sequences have in eliminating false predictions caused by character sequences.

As shown in Fig 4I–4L, FastText achieves the best performance among the four base learners when using only character sequences as input. However, when only radical sequences are used, the accuracy generated by FastText is significantly lower than that of TextCNN and BI-GRU. Overall, no negative effect occurs when combining radical features with character features; in contrast, the accuracy increases by 0.003. Only the "Home Furniture" and "Finance" classes undergo a minor reduction in accuracy.

## Sentiment-based text classification experiment

Next, we conducted another experiment to analyze the importance of radical features in the sentiment-based Chinese classification task. The results are shown in Table 5.

As Table 5 shows, the accuracy rankings of the four base learners are the same as those on the domain-based Chinese text classification task. FastText is better than BI-GRU, and BI-GRU is better than TextCNN and DNN. An interesting fact is that the accuracies of FastText and BI-GRU with pure radical features are 0.731 and 0.744, respectively, and both are higher than those of TextCNN and DNN using character features as input. Similarly, although the accuracies using pure radical features as input are quite different, they all improve when radical features are included: DNN, TextCNN, BI-GRU, and FastText improve by 0.023, 0.064, 0.023 and 0.011, respectively. Clearly, the radical features help to enhance the performance of sentiment-based Chinese classification. The fine-grained improvement from including radical features in each base learner is shown in Fig 5.

The three confusion matrices in Fig 5A–5C were obtained from DNN given character features, radical features and combined features as input. Intuitively, the accuracy of the "Positive" category is higher than that of the "Negative" category, and that of the "Negative" category is higher than that of the "Neutral" category. Comparing Fig 5A with Fig 5C, the

**Table 5. Base learner accuracy on the sentiment-based text classification.**

| Number | base learner | accuracy | | |
|---|---|---|---|---|
| | | character only | radical only | combined |
| 1 | DNN | 0.716 | 0.638 | 0.739 |
| 2 | TextCNN | 0.725 | 0.703 | 0.789 |
| 3 | BI-GRU | 0.743 | 0.731 | 0.776 |
| 4 | FastText | 0.783 | 0.744 | 0.794 |

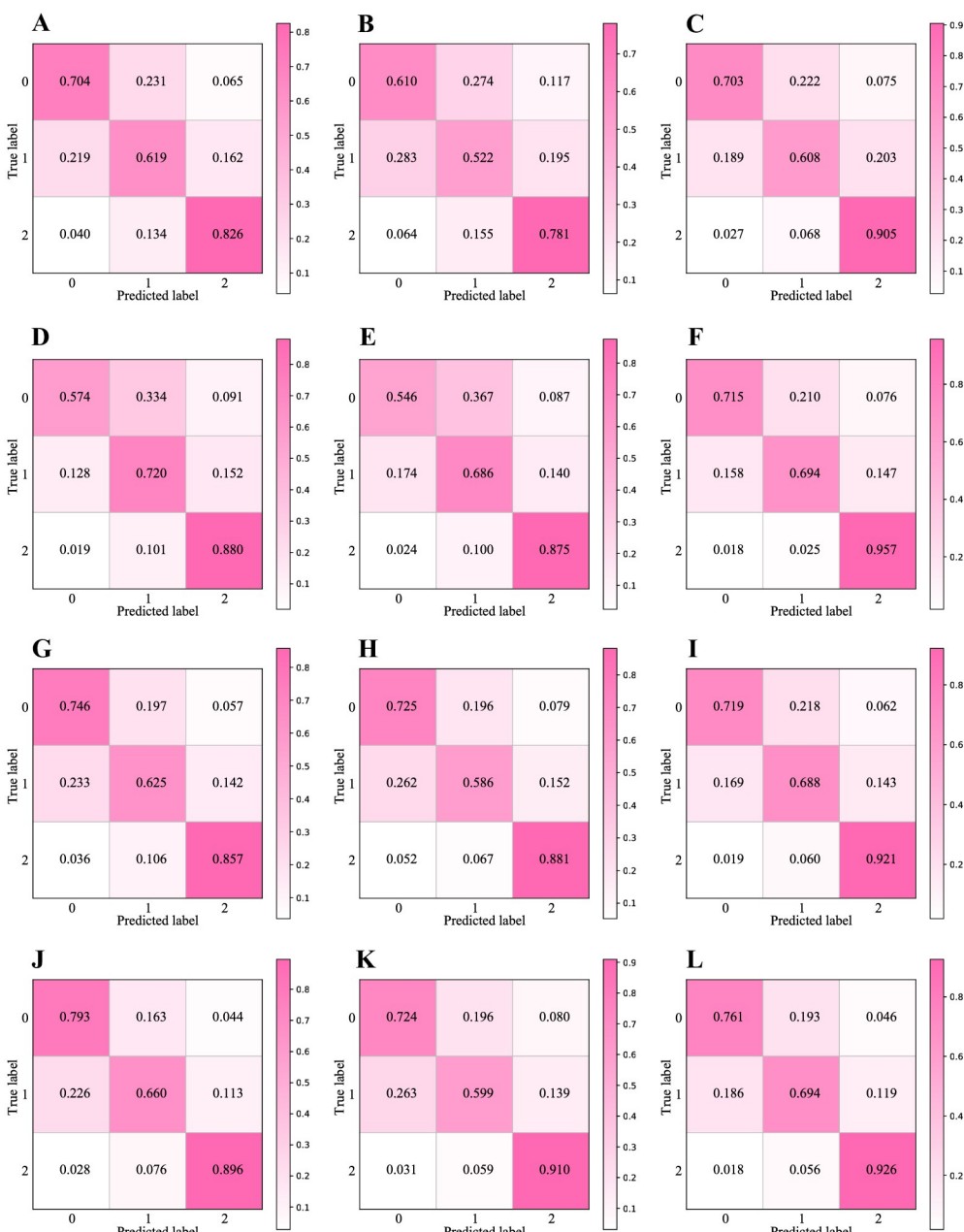

**Fig 5. Sentiment-based classification confusion matrices.** The results for different methods are shown: DNN with (A) character, (B) radical, (C) combined features; TextCNN with (D) character, (E) radical, (F) combined features; BI-GRU with (G) character, (H) radical, (I) combined features; FastText with (J) character, (K) radical, (L) combined features. 0-Negative, 1-Neutral, 2-Positive.

accuracy of "Positive" improves by 0.079, but the accuracy of "Negative" falls by 0.001, while the accuracy of "Neutral" falls by 0.011, leading to an overall improvement of only 0.023. That is, adding the radical features enhances the performance of the DNN in identifying positive semantic information, but its ability to recognize neutral semantic information is reduced. Fig 5D–5F show the accuracies and error rates of the TextCNN. First, the CNN model tends to identify "Positive" samples, and the accuracies of "Positive" in matrices Fig 5D–5F are all

higher than those of the other categories. In matrices Fig 5D and 5E, the accuracy of the "Neutral" sentiment is higher than that of the "Negative" sentiment. However, when we combine the character and radical features, the generated matrix Fig 5F is quite different from Fig 5D and 5E. The accuracy for "Positive" and "Negative" sentiments improves further to 0.957 and 0.715, respectively. Meanwhile, the accuracy for "Neutral" sentiment decreases to 0.694. Approximately 15.8% and 14.7% of "Neutral" samples are mispredicted as "Negative" and "Positive", respectively.

Fig 5G–5I show the BI-GRU results. Overall, the accuracies on the "Positive" category are higher than those on the "Negative", which in turn are higher than those of the "Neutral" category. The accuracies on the "Positive" and "Neutral" categories increase by 0.064 and 0.063, respectively, after adding the radical features, while the "Positive" category accuracy decreases by 0.027.

Fig 5J–5L shows that FastText is similar to BI-GRU; it performs better on the Positive category than on the "Negative" category and that the "Negative" category accuracy is better than its accuracy on the "Neutral" category. After adding the radical features, the "Positive" accuracy increases by 0.030, the "Neutral" accuracy increases by 0.034, and the "Negative" accuracy decreases by 0.031.

Considering the above, in both domain-based and sentiment-based classification tasks, although the performance of radical sequence varies in different base learners, most combined sequences yielded higher accuracy compared to character sequences, indicating that radical features can benefit Chinese text classification tasks. More importantly, pure radical sequences achieved decent performance overall and were sometimes equivalent to character sequences.

## RQ4: Analyzing the importance of base learners and finding the most efficient ensemble setup

### Base learner ablation

In this section, we further analyze the importance of the four base learners to the entire model. We conducted two groups of ablation experiments to observe the accuracy changes after removing one or two learners as shown in Fig 6.

The first is the domain-based Chinese text classification experiment. As shown in Fig 6A, we found that after removing the DNN, TextCNN or BI-GRU, the final accuracies remained stable at 0.937; however, after removing FastText, the accuracy fell to 0.965, a reduction of 0.007. This shows that among the four base learners, FastText has the highest weight because the overall accuracy is reduced when it is removed. The other three learners, however, are not critical in this group of experiments because the overall accuracy does not change when they are removed.

In the groups with two base learners, we found that the accuracies of all the tasks with FastText engaged exceeded 0.97. In particular, the combination of FastTex with TextCNN reached 0.973, while in the tasks without FastText, the accuracies were all below 0.97.

The classification results on the sentiment-based corpus are shown in Fig 6B. Similar to the domain classification, after removing FastText, the accuracy decreased to 0.789, while after removing TextCNN and BI-GRU, only a slight decline occurred. One interesting phenomenon is that after removing the DNN, the accuracy increased by 0.001, which indicates that the probability vectors produced by the DNN interfere with XGBoost. In addition, in the tasks with pairwise base learners, the accuracy of FastText combined with TextCNN reached 0.806, that of BI-GRU reached 0.800, and those of the other combinations were all less than 0.800.

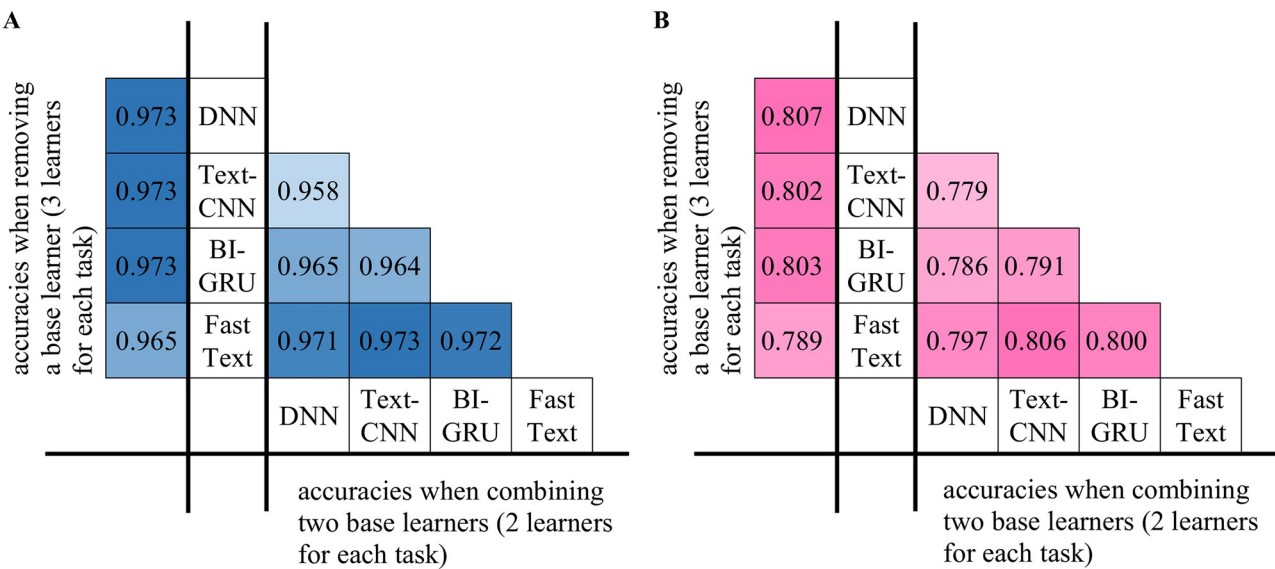

**Fig 6. Overall accuracy scores with different combinations of base learners on two classification tasks.** A Domain classification. B: Sentiment classification.

Both groups of experiments indicate the importance of FastText for Chinese text classification. The combination of TextCNN and FastText achieves performance equal to that when all four learners are combined. In the domain classification task, removing any other base learners except FastText does not reduce the accuracy. However, the DNN has a slight negative effect on sentiment-based text classification. Beyond the DNN, removing any type of base learner leads to different degrees of accuracy reduction.

## Efficiency analysis

Considering that the base learners have different architectures, the training time required by each learner to fit the training samples varies. Therefore, we analyzed the efficiency from a practical application perspective. During the training stage, the average learning times of the DNN, TextCNN and FastText are within several seconds of each other, while the BI-GRU model needs hundreds of seconds due to its sequential structure. In our experiments, DNN, TextCNN and FastText converge after 10 epochs, whereas BI-GRU requires only 5 epochs. Therefore, based on the average learning times and the accuracies of different combinations, we drew correlation analysis figures for the two groups of experiments, as shown in Fig 7.

Again, we find that the tasks that include the BI-GRU are more time-consuming. As mentioned above, the TextCNN and FastText combination has equivalent accuracy to the entire model but requires less time. The accuracy of 0.973 achieved by TextCNN and FastText ranks first in the domain-based experiments, and that combination's accuracy of 0.806 is only one thousandth lower than that of the highest in sentiment-based experiments. However, the overall average learning times for the TextCNN and FastText combination are only hundreds of seconds for the two tasks, respectively, or only 3% of the highest model combination. Therefore, we believe that in actual applications of Chinese classification, the XGBoost-based ensemble learning model with TextCNN and FastText as base learners is the most effective and efficient.

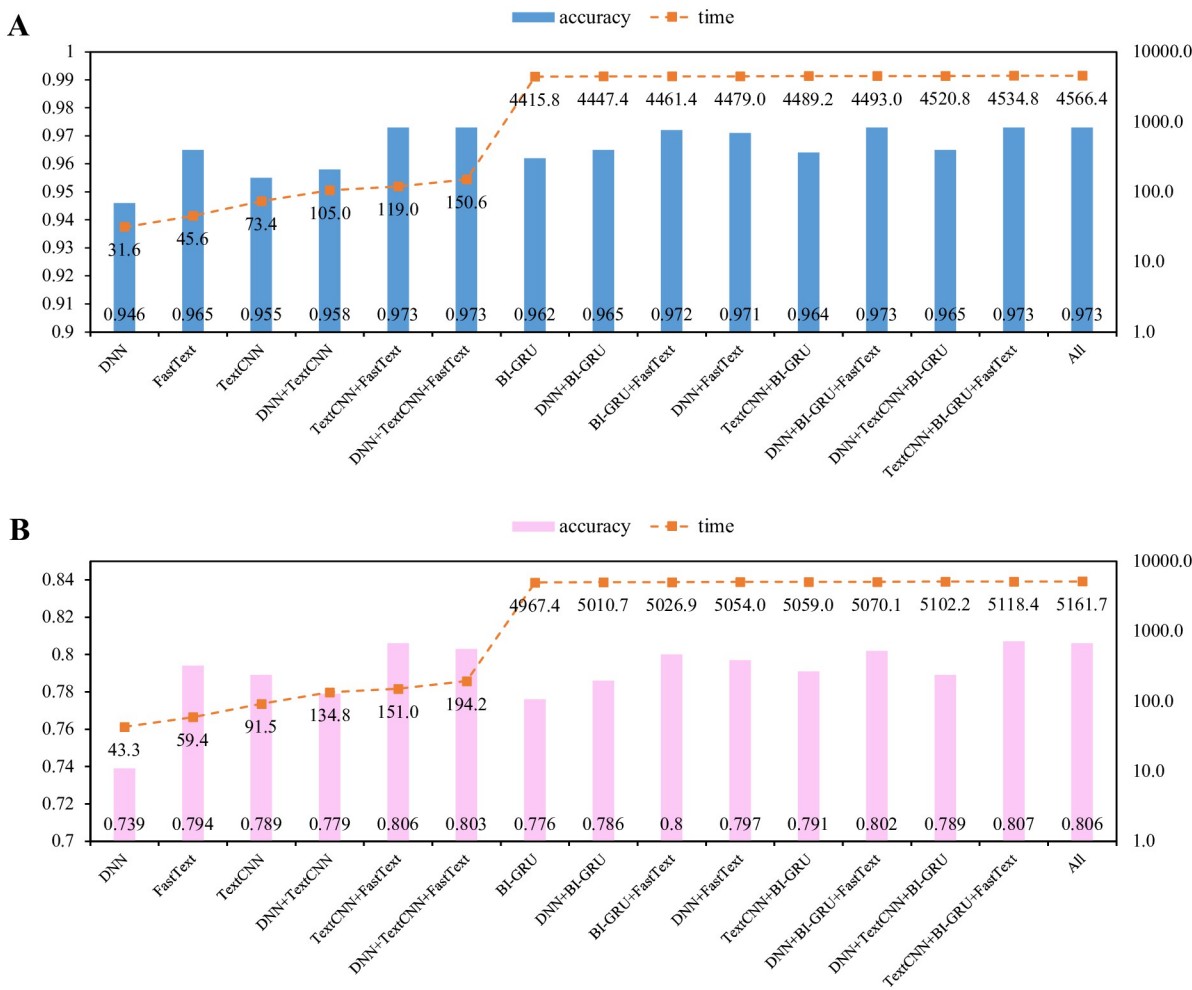

**Fig 7. Training times and accuracies on the two classification tasks.** A: Domain classification. B: Sentiment classification.

## Discussion

### Findings

An ensemble learning method for Glyph-aware Chinese text classification (LEGACT) is proposed. To verify the effectiveness and efficiency of the proposed model, we conducted several experiments with a domain-based dataset and a sentiment-based dataset. First, the proposed method yields superior performance compared with all the state-of-the-art deep networks whilst containing fewer trainable parameters and requiring less computational capacity. The experimental result can support that our ensemble method combining two types of features is an effective and lightweight solution for glyph-aware Chinese classification tasks.

In addition, it is notable that even when provided with only traditional character features, some shallow networks with a voter can outperform many models with deep network architecture on Chinese text classification tasks. In section 5, we conducted a fine-grained comparison and analysis, which revealed that the radical features helped to improve the classification performance: all the base learners achieved better accuracies when radical features were included in their input. Despite the total number of radical token features being less than 300, all the

learners performed well on both the 10-category domain corpus and the 3-category sentiment corpus.

Section 6 analyzed the weight of each base learner. FastText achieves the highest importance among the four base learners. More importantly, it does not require much training time or computation capacity. Finally, we find that the TextCNN and FastText combination not only achieves an effect equivalent to that of using all four base models in the overall method but also saves considerable training time and hardware resources.

## Implications

This study has several methodological and practical implications.

**Pure radical features can achieve high levels of accuracy**. A significant number of studies have shown the benefits of glyph features in Chinese NLP tasks. In this paper, we not only detailed the impact of glyph features on accuracy scores in domain and sentiment classification tasks, but also concluded that pure radical features can achieve high levels of accuracy. Previous research [37] has shown that word segmentation is not necessary in deep learning-based Chinese NLP, and that using only character tokens produces equivalent performance. In this study, we further investigated the use of radical tokens for classification tasks and found that radical features can perform the task with decent accuracy. Given that the number of radical tokens is only around 200, while the number of common Chinese characters is over 3,000, using only radical tokens can be a viable option when computational resources are limited and high performance is unnecessary.

**An ensemble model consisting of TextCNN and FastText as base learners and XGBoost as a meta-learner is the most efficient solution**. In practice, given that Chinese is one of the most widely spoken languages and that there is an abundance of Chinese text available on the internet, it is crucial to develop and improve language representation and processing methods for Chinese. The method presented in this paper has shown to be an effective solution compared to many baselines, providing a lightweight solution for Chinese NLP research, especially when computational resources are limited. Furthermore, for text classification tasks, we found that our method (using TextCNN and FastText as base learners and XGBoost as a meta-learner) is the most efficient solution, making it suitable for widespread use in Chinese classification systems.

**A shallow ensemble can produce equivalent or even superior performance compared to the integration of deeper models**. As deep learning techniques and computer hardware continue to advance, large-scale deep learning models have become increasingly popular and have shown to be beneficial for many prediction tasks [52]. However, due to the computational resources required, research institutions or scholars with access to powerful computing machines have an advantage over those without [53, 54]. This may make it challenging for institutions or scholars without access to these resources to keep up with the latest research developments and to compete in the field. In this paper, we have demonstrated, at least in the case of Chinese text classification tasks, that properly assembling several shallow networks can produce performance comparable to that of large-scale models. Therefore, we call for further research to follow this research paradigm on specific prediction tasks and to develop more computationally efficient deep learning models.

## Conclusion

To balance the trade-off between model complexity and performance for glyph-aware Chinese text classification tasks, we proposed a lightweight ensemble solution that uses several shallow networks as base learners and Xgboost as a meta-learner. Through a series of experiments, we

have shown the efficiency and effectiveness of this approach. This study also has some limitations. Firstly, only the two most popular text classification scenarios (domain classification and sentiment classification) are considered, while other fine-grained scenarios are ignored due to space constraints. Secondly, all the modules within the proposed approach are existing models; therefore, the innovation of this study is limited. In future work, we plan to optimize the proposed method and apply it to other fields.

## Acknowledgments

Both authors thank the journal reviewers and editor for their helpful suggestions.

## Author Contributions

**Conceptualization:** Ping Wang.

**Data curation:** Jingrui Hou.

**Formal analysis:** Ping Wang.

**Funding acquisition:** Ping Wang.

**Investigation:** Jingrui Hou.

**Methodology:** Jingrui Hou.

**Project administration:** Ping Wang.

**Resources:** Jingrui Hou, Ping Wang.

**Software:** Jingrui Hou.

**Supervision:** Ping Wang.

**Validation:** Jingrui Hou.

**Visualization:** Jingrui Hou.

**Writing – original draft:** Jingrui Hou, Ping Wang.

**Writing – review & editing:** Jingrui Hou, Ping Wang.

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
