## [Decision Letter · Decision Letter 0]

1 Jun 2023

PONE-D-23-09290Assemble the Shallow or Integrate a Deep? Toward a Lightweight Solution for Glyph-aware Chinese Text ClassificationPLOS ONE

Dear Dr. Wang,

Thank you for submitting your manuscript to PLOS ONE. After careful consideration, we feel that it has merit but does not fully meet PLOS ONE’s publication criteria as it currently stands. Therefore, we invite you to submit a revised version of the manuscript that addresses the points raised during the review process.

We look forward to receiving your revised manuscript.

Kind regards,

Abel C.H. Chen

Academic Editor

PLOS ONE

Journal Requirements:

   "This work was supported by the National Natural Science Foundation of China [No.72074171] and the Chinese Scholarship Council (CSC) [No. 202208060371]."

   ""NO""

   "NO"

7. Please ensure that you refer to Figure 7 in your text as, if accepted, production will need this reference to link the reader to the figure.

Reviewers' comments:

Reviewer's Responses to Questions

**Comments to the Author**

1. Is the manuscript technically sound, and do the data support the conclusions?

Reviewer #1: Yes

2. Has the statistical analysis been performed appropriately and rigorously? 

Reviewer #1: Yes

3. Have the authors made all data underlying the findings in their manuscript fully available?

Reviewer #1: Yes

4. Is the manuscript presented in an intelligible fashion and written in standard English?

Reviewer #1: Yes

5. Review Comments to the Author

Reviewer #1: 1.The authors propose an effective solution to the problem that most current research does not consider internal features of Chinese text but focuses on complex deep network architectures.

2.By reading “Base Learning”, the authors summarized in one sentence that DNN,Text-CNN and BI-GRU are used to encode character features and radical features, which is a bit common. In addition, the description of Figure 1 is too little, and I think authors should briefly describe 4 components in the figure.

3.A proper name can be given to the lightweight Chinese character text classification method proposed in this paper.

4.Experiments are well covered, with comparison experiments, ablation experiments, and efficiency analysis. The result diagrams are good and easy to understand.

5.The experimental section has a large number of confusion matrices; their layout needs to be improved.

6. PLOS authors have the option to publish the peer review history of their article (what does this mean?). If published, this will include your full peer review and any attached files.

Reviewer #1: **Yes: **Jin Liu

---

## [Author Response · Author response to Decision Letter 0]

17 Jun 2023

Response to the editor

Comment 1: Please ensure that your manuscript meets PLOS ONE’s style requirements, including those for file naming. The PLOS ONE style templates can be found at

Our response: We appreciate your guidance in directing us towards the PLOS ONE style templates. We have diligently applied these templates to refine and reformat our manuscript, ensuring it adheres to the style requirements of PLOS ONE. Specifically, we have made the following modifications:

1) The title has been revised to comply with the sentence case rule: “Assemble the shallow or integrate a deep? Toward a lightweight solution for glyph-aware Chinese text classification.”

2) The abbreviation "UK" has been expanded to its full form, "United Kingdom."

3) We have separated the corresponding author's statement and their email address into two distinct lines. Following this, we have included the corresponding author's initials in parentheses after the email address.

4) We have adjusted the font sizes for the headings: Level 1 headings are now in 18pt, bold type, and Level 2 headings are in 16pt, bold type.

5) We have standardized the citation format for figures from "Figure x" to "Fig x."

6) All figure captions have been reformatted to appear in bold.

7) We have reformatted the paragraphs to a double-spaced layout.

Comment 2: Please note that PLOS ONE has specific guidelines on code sharing for submissions in which author-generated code underpins the findings in the manuscript. In these cases, all author-generated code must be made available without restrictions upon publication of the work. Please review our guidelines at https://journals.plos.org/plosone/s/materials-and-software-sharing#loc-sharing-code and ensure that your code is shared in a way that follows best practice and facilitates reproducibility and reuse.

Our response: We appreciate your guidance regarding the PLOS ONE's policy on code sharing. In accordance with these guidelines, we plan to make all author-generated code integral to our study freely available to ensure reproducibility and facilitate reuse. To achieve this, we will publish our code, along with a comprehensive tutorial, on a GitHub repository - a platform renowned and widely utilized within the computer science research community.

Comment 3: Thank you for stating the following in the Acknowledgments Section of your manuscript:

“This work was supported by the National Natural Science Foundation of China [No.72074171] and the Chinese Scholarship Council (CSC) [No. 202208060371].”

We note that you have provided funding information that is not currently declared in your Funding Statement. However, funding information should not appear in the Acknowledgments section or other areas of your manuscript. We will only publish funding information present in the Funding Statement section of the online submission form. Please remove any funding-related text from the manuscript and let us know how you would like to update your Funding Statement. Currently, your Funding Statement reads as follows: ““NO””

Our response: Thank you for your guidance regarding the funding statement. We have now removed the funding information from the acknowledgment section in the manuscript. The updated acknowledgment section reads as follows: "Jingrui Hou is a PhD student supported by the China Scholarship Council and Loughborough University. Both authors extend their gratitude to the journal reviewers and editor for their valuable suggestions."

We have already added the founding information in the submission system, shown as “National Natural Science Foundation of China; Award Number: 72074171 | Recipient: Ping Wang”. 

Comment 4: We note that the grant information you provided in the ‘Funding Information’ and ‘Financial Disclosure’ sections do not match.

Our response: Thanks for your kind reminder. We have thoroughly rechecked our funding information in line with your recommendation to ensure complete accuracy.

Comment 5: Thank you for stating the following financial disclosure: “NO”

Please state what role the funders took in the study. If the funders had no role, please state: “The funders had no role in study design, data collection and analysis, decision to publish, or preparation of the manuscript.”

Our response: We greatly appreciate your guidance on the matter. We have now updated our cover letter to include the additional funding information and statement as follows: "This work was financially supported by the National Natural Science Foundation of China, under grant number 72074171. The funder did not participate in the study design, data collection and analysis, decision to publish, or in the preparation of the manuscript." We hope that this update aligns with your instructions. Thank you once again for your assistance.

Comment 6: PLOS requires an ORCID iD for the corresponding author in Editorial Manager on papers submitted after December 6th, 2016. Please ensure that you have an ORCID iD and that it is validated in Editorial Manager. To do this, go to ‘Update my Information’ (in the upper left-hand corner of the main menu), and click on the Fetch/Validate link next to the ORCID field. This will take you to the ORCID site and allow you to create a new iD or authenticate a pre-existing iD in Editorial Manager. Please see the following video for instructions on linking an ORCID iD to your Editorial Manager account: https://www.youtube.com/watch?v=_xcclfuvtxQ

Our response: We appreciate your advice on linking ORCID with the Editorial Manager account. Both authors have taken steps to accomplish this.

Comment 7: Please ensure that you refer to Figure 7 in your text as, if accepted, production will need this reference to link the reader to the figure.

Our response: We appreciate your diligence in ensuring that all figures are appropriately referenced within the text. Following your comment, we have now incorporated a reference to Figure 7 within the body of our manuscript. Furthermore, we have meticulously reviewed all other figures and tables to ensure they are correctly cited.

Comment 8: Please review your reference list to ensure that it is complete and correct. If you have cited papers that have been retracted, please include the rationale for doing so in the manuscript text, or remove these references and replace them with relevant current references. Any changes to the reference list should be mentioned in the rebuttal letter that accompanies your revised manuscript. If you need to cite a retracted article, indicate the article’s retracted status in the References list and also include a citation and full reference for the retraction notice.

Our response: We greatly appreciate your guidance on refining our reference list. We have conducted a meticulous review and made the necessary amendments:

1) We have corrected the author information for reference [46] to: “Jinhuakst. Chinese NLP Corpus; 2017.https://github.com/SophonPlus/ChineseNlpCorpus.”

2) We have removed the invalid DOI information for reference [38], which now reads: “Ren J, Cui J, Shah M, Ryu JT, Kwon D. A study on comparison analysis of the DNN, CNN, and RNN models for network anomaly detection. EEO. 2020;19(4):947–956.”

3) We have supplemented citation [11] with detailed information: “Meng Y, Wu W, Wang F, Li X, Nie P, Yin F, et al. Glyce: Glyph-vectors for Chinese character representations. In: Proceedings of the 33rd International Conference on Neural Information Processing Systems. Red Hook, NY, USA: Curran Associates Inc.; 2019.”

4) To enhance accessibility, we have added the DOI link prefix (https://doi.org/) for several references ([1], [15]-17], [19-23], [25-27], [32-36], [44], [51-53]) to make their associated URLs functional.

We have not cited any retracted articles in our manuscript. Thank you once again for your valuable assistance in ensuring the accuracy and completeness of our reference list.

Response to the reviewer comments

Comment 1: The authors propose an effective solution to the problem that most current research does not consider internal features of Chinese text but focuses on complex deep network architectures.

Our response: We truly appreciate your valuable feedback and recognition of our work. Indeed, the primary aim of our research is to present a lightweight yet effective Glyph-aware Chinese Text Classification solution.

Our proposed system, comprising various modules (DNN, Text-CNN, Bi-GRU, FastText, Xgboost), maintains a modular design where the first four sub-modules operate independently during training. This modular design ensures flexibility, allowing the system to be expanded or compressed as needed, making it highly accommodating in scenarios with limited computational resources.

Additionally, we have incorporated Xgboost, a traditional machine learning algorithm, which does not necessitate additional GPU overhead. Further, to optimize our model, we conducted ablation experiments, successfully pruning the model and discovering an equally effective model subset (Text-CNN, FastText, and Xgboost) that performs on par with the full original model.

In light of your suggestion, we will continue to refine our approach, striving to develop a model that remains lightweight while effectively leveraging the intrinsic features of Chinese text. Thank you once again for your insightful input.

Comment 2: By reading “Base Learning”, the authors summarized in one sentence that DNN,Text-CNN and BI-GRU are used to encode character features and radical features, which is a bit common. In addition, the description of Figure 1 is too little, and I think authors should briefly describe 4 components in the figure.

Our response: We appreciate your valuable feedback, which provides an opportunity to further clarify our methodology and graphical representation.

Regarding your first point on the "Base Learning" section, we concur that our summary sentence regarding the use of DNN, Text-CNN, and BI-GRU to encode character and radical features might have been somewhat generic. However, our rationale for choosing these particular architectures as independent encoders stems from their fundamental role in deep learning. One of our primary research goals is to demonstrate the superiority of an ensemble of shallow networks over more intricate models. For a more in-depth understanding of our choice of these four modules, we refer readers to the first subsection of Section 3.2 where this is discussed in detail.

Regarding your second point about Figure 1, we have responded by enriching the figure caption and extending the descriptions of the four components within the body of the manuscript. The caption is “The depicted Deep Neural Network (DNN) comprises a hidden layer bridging the input and output layers. It utilizes a Global Average Pooling layer for encoding tensor dimensionality reduction. The Text-CNN is structured with three sets of convolution and max-pooling layers for input vector encoding, and its output is subsequently flattened into a one-dimensional vector purposed for classification. The Bidirectional Gated Recurrent Unit (BI-GRU) is a sequential model featuring two GRU layers operating in forward and backward directions. FastText arranges the input into uni-gram, bi-gram, and tri-gram features, and the resulting embedding vectors are introduced into a global average pooling layer.”

We also improved the description of this approach in the manuscript. It is: “DNN, Text-CNN, and GRU utilize a common model input and embedding. The DNN segment applies several hidden layers, connected by fully connected layers, to the character or radical level embedding matrix, followed by a global max pooling layer to normalize sequence lengths and extract the maximal value from the hidden layer output. The Text-CNN processes the embedding matrix through a cyclical operation that consists of a one-dimensional convolution layer and a max pooling layer. The convolution layer extracts semantic features using a sliding convolution kernel, while the max pooling layer streamlines parameter size and ensures a constant-length input for subsequent cycles. Post-convolution and pooling, the two-dimensional vectors are flattened, with a batch normalization layer added for smoothing. The Bi-GRU structure learns the semantic representation by considering forward and backward text sequences. The final representation is formed by the concatenation of two semantic encodings obtained after updating all sequence elements. FastText, though possessing a simplistic network structure, augments the input with n-gram features. These features, inclusive of uni-, bi-, and trigram aspects, are integrated to create a new sequence. This sequence is embedded and directed to the global average pooling layer to enhance sentence representation.”

The following content is removed: “The DNN has only one hidden layer that connects the input and output layers. A global average pooling layer is used to reduce the dimensionality of the encoding tensor. The Text-CNN is composed of three groups of convolution and max pooling layers to encode the input vectors. The output is then flattened to generate a one-dimensional vector for classification. The BI-GRU is a sequential model with two connected GRU layers, one in the forward direction and one in the backward direction. FastText reorganizes the input as uni-gram, bi-gram, and tri-gram features, and their embedding vectors are put into a global average pooling layer. The output is then directly used for classification.”

Comment 3: A proper name can be given to the lightweight Chinese character text classification method proposed in this paper.

Our response: We appreciate your thoughtful suggestion to assign a distinctive name to the lightweight Chinese character text classification method we propose in this paper. After a careful deliberation, we have chosen to designate the proposed approach as “Light Ensemble for Glyph Aware Chinese Text Classification,” abbreviated to “LEGACT.”

To integrate this change into our manuscript, we have updated the relevant sentences as follows:

1) In the Abstract section, we have revised the sentence to read: “To tackle this challenge, we introduce a lightweight ensemble learning method, Light Ensemble for Glyph Aware Chinese Text Classification (LEGACT). This method unites typical shallow networks as base learners with machine learning classifiers serving as meta-learners.”

2) In the third section, we have updated the sentence to read: “In this section, we provide a detailed outline for constructing our novel Light Ensemble for Glyph Aware Chinese Text Classification model (LEGACT).”

3) The sentence in the Discussion section has been modified to: “We present a novel ensemble learning method, termed Light Ensemble for Glyph Aware Chinese Text Classification (LEGACT).”

4) The captions of Fig 1 and Fig 2, as well as the content of Table 3, have been appropriately amended to incorporate this name.

Comment 4: Experiments are well covered, with comparison experiments, ablation experiments, and efficiency analysis. The result diagrams are good and easy to understand.

Our response: We sincerely appreciate your positive remarks of our experimental approach. Your feedback greatly validates our efforts and plays a significant role in enhancing the overall quality of our article. Thank you once again for your constructive and encouraging comments.

Comment 5. The experimental section has a large number of confusion matrices; their layout needs to be improved.

Our response: Thank you for your insightful feedback. In the original version, we did indeed employ a singular large diagram to represent all matrices and relied on manual typesetting, which may have resulted in less optimal layout. In our revised manuscript, we have restructured the presentation of our data. We now utilize a series of smaller, individually illustrated diagrams. These have been organized via an automated typesetting framework, enhancing readability and visual appeal. Please refer to the updated Fig 4 and Fig 5 for these changes.

---

## [Decision Letter · Decision Letter 1]

14 Jul 2023

Assemble the shallow or integrate a deep? Toward a lightweight solution for glyph-aware Chinese text classification

PONE-D-23-09290R1

Dear Dr. Wang,

We’re pleased to inform you that your manuscript has been judged scientifically suitable for publication and will be formally accepted for publication once it meets all outstanding technical requirements.

Kind regards,

Abel C.H. Chen

Academic Editor

PLOS ONE

Additional Editor Comments (optional):

Reviewers' comments:

Reviewer's Responses to Questions

**Comments to the Author**

1. If the authors have adequately addressed your comments raised in a previous round of review and you feel that this manuscript is now acceptable for publication, you may indicate that here to bypass the “Comments to the Author” section, enter your conflict of interest statement in the “Confidential to Editor” section, and submit your "Accept" recommendation.

Reviewer #1: All comments have been addressed

2. Is the manuscript technically sound, and do the data support the conclusions?

Reviewer #1: Yes

3. Has the statistical analysis been performed appropriately and rigorously? 

Reviewer #1: Yes

4. Have the authors made all data underlying the findings in their manuscript fully available?

Reviewer #1: Yes

5. Is the manuscript presented in an intelligible fashion and written in standard English?

Reviewer #1: Yes

6. Review Comments to the Author

Reviewer #1: The authors consider the internal features of Chinese text and propose a lightweight but powerful glyph-aware Chinese text classification solution model, and demonstrate the effectiveness of the model on the public datasets THUCNews and Yf_amazon.

In response to a previous review suggestion, the authors named the model a light weight ensemble learning method for glyph-aware Chinese text classification (LEGACT). Secondly, the authors have added more details to the illustrations, e.g., the four components of the Base learner architecture in LEGACT (Fig. 1.), which are described accordingly to make the paper more convincing. Then, the authors rearranged the experimental results to make the layout more attractive. Finally, the authors have corrected some minor grammatical errors and terminology in the paper.

Overall I would say the paper is acceptable.

7. PLOS authors have the option to publish the peer review history of their article (what does this mean?). If published, this will include your full peer review and any attached files.

Reviewer #1: No

---

## [Editor Report · Acceptance letter]

19 Jul 2023

PONE-D-23-09290R1 

Assemble the shallow or integrate a deep? Toward a lightweight solution for glyph-aware Chinese text classification 

Dear Dr. Wang:

I'm pleased to inform you that your manuscript has been deemed suitable for publication in PLOS ONE. Congratulations! Your manuscript is now with our production department. 

Kind regards, 

on behalf of

Dr. Abel C.H. Chen 

Academic Editor

PLOS ONE